# Unmasking social distant damage of developed regions' lifestyle: A decoupling analysis of the indecent labour footprint

**Ángela García-Alaminos**[¤]*, **Fabio Monsalve**[¤], **Jorge Zafrilla**[¤], **Maria-Angeles Cadarso**[¤]

Department of Economic Analysis and Finances, Global Energy and Environmental Economics Analysis Research Group, University of Castilla-La Mancha, Albacete, Spain

¤ Current address: Faculty of Economics and Business, University of Castilla-La Mancha, Albacete, Spain
* angela.garcia@uclm.es

**Data Availability Statement:** The WIOD data in its 2016 release is available at www.wiod.org: Timmer MP, Dietzenbacher E, Los B, Stehrer R, de Vries GJ. An Illustrated User Guide to the World Input–

## Abstract

A fair path to achieve a sustainable world would imply reducing the eventual negative effects linked to the production process while increasing economic output, which is referred to in the literature as impact decoupling. This article aims to assess whether global consumption chains are currently on the decoupling path or not, from a social point of view. Specifically, we address the working conditions which developed societies' lifestyle sparked at a distance in global factory countries, focusing on the most harmful consequences of an indecent work. Additionally, we determine the kind of decoupling observed through the new concept of social footprints' elasticities with respect to final demand for each region. We employ a Multi-Regional Input-Output model and an own elaboration database of social impacts concerning undignified working conditions. Results indicate that most countries achieved the goal of decoupling occupational injuries -both fatal and non-fatal- from production, while results for forced labour show a slower and sometimes uncertain process of decoupling. European Union and United States' footprints have been reduced overtime for the three impacts. However, more than half of these footprints are still generated by imports, mainly from developing regions.

## Introduction

Peter Singer alerts, in his widely read *Practical Ethics* [1], that our moral thinking is shaped by the development of principles which help us deal with problems within our community, not those outside it. Therefore, we lack any kind of instinctive inhibitions or emotional responses against the imperceptible new ways in which we can harm one another. Singer illustrates this moral flaw with the example of the releasing of waste gases, but the same logic of "imperceptible distant damage" operates in the social sphere. These kinds of distant harms challenge humanity's current major agenda; i.e., the Sustainable Development Goals (SDG) [2]. If we are not aware of our potentially destructive consumption habits due to imperceptibility and distance, it will become more difficult to achieve those ambitious goals. On the other hand, such heterogeneous targets drive several dichotomies, trade-offs and synergies [3, 4] along with

Output Database: the Case of Global Automotive Production. Review of International Economics. 2015;23(3):575-605. https://doi.org/10.1111/roie.12178. The satellite social account is available at https://data.mendeley.com/datasets/6h5msdfjk2/1 under license CC BY 4.0: García-Alaminos A. Social Indicators of Working Conditions Database. 1 ed. Mendeley Data2019. http://dx.doi.org/10.17632/6h5msdfjk2.1. The MATLAB code used in this text to perform the MRIO analysis is available directly from the authors on request.

**Funding:** F.M., J.Z. and M.-A.C thank the Ministry of Economy and Competitiveness of Spain for funding the research project "ECO2012-33341 that led to this paper. Á. G-A. also thanks the European Social Fund and University of Castilla-La Mancha support through the Regional FPI program (PRE-18: Contratos predoctorales para la formación de personal investigador en el marco del plan propio de I+D+i, susceptible de cofinanciación por el Fondo Social Europeo). All the authors thank "Plan Propio de Investigación" from University of Castilla-La Mancha (reference 2019-GRIN-27108) for funding this article. The funders had no role in study design, data collection and analysis, decision to publish, or preparation of the manuscript.

**Competing interests:** The authors have declared that no competing interests exist.

their achievement. Furthermore, global value chains and relocation of production may counteract the decoupling of negative impacts from economic growth [5].

Focusing on the labour dimension of social progress, the SDG number 8 urges nations to promote decent work for all, understanding it as "work that is productive and delivers a fair income, security in the workplace and social protection for families, better prospects for personal development and social integration"[6]. Literature has shown that development attained through openness to international trade does not always carry an improvement to working conditions [7]. Mosley and Uno [8] claim that the quality of labour rights inside a region is negatively correlated with its degree of openness in international trade after controlling domestic variables like level of democracy, among others. In fact, huge flows of bad work embodied in trade are arising from developing regions to wealthy nations [9], creating a network in which servant countries support lifestyle of master coutries [10]. The most abject phenomena like labour slavery are still a reality all over the world, not only in developing countries in Asia, South America or Africa, where even children are involved in forced labour [11, 12], but also in developed regions such as Europe [13], especially affecting the most vulnerable groups–migrants and refugees—in host countries with weak institutions [14]. In addition, the relationship between safety at work and development is also a controversial binomial. Occupational injuries and fatalities are also a great concern when evaluating worldwide supply chains [15]. Some emerging countries seem to flaunt a higher tolerance towards its citizens' health deterioration as a competitive advantage [16] and in some cases, a "race to the bottom" in labour protection and labour rights is found, where countries competitively undercut each other's labour standards [17, 18]. Extending the social impact of work, in a broad sense, through a synthetic indicator, Alsamawi, Murray [19] claim that countries tend to import more negative impacts embodied in traded products as they reach higher development levels. Therefore, the efforts to achieve environmentally-friendly and socially-desirable development seems to be displaced "at distance" as responsibilities can be transferred to "less concerned" regions.

In the aforementioned social sustainability context in which labour standards play an essential role, the main aim of this article is the evaluation of indecent labour patterns in terms of decoupling from a footprint perspective and the introduction of the concept of social elasticity.

Three main elements constitute the basis of our analysis. First, indecent labour practices are measured trought three indicators retrieved from an own-elaboration multiregional dataset: non-fatal injuries, fatal injuries and forced labour (for further information about these indicators, please check S2 Appendix). Despite the relevance of having a decent job is a decisive element in any population's quality of life, usual employment accounts seem to be insufficient to reflect all qualitative dimensions that constitute the idea of a proper job. According to Stiglitz, Sen [20], the gap between economic growth and perceived well-being by the population seems to be increasing, which creates an imperative to develop reliable social indicators that complement the current measurements to draw up a more humane concept of development. The three variables proposed in this article intend to quantify the most harmful dimensions of indecent labour, which are those endangering the physical and moral integrity of workers. The worst hazards that can derive from indecent labour are either suffering some health damage–even death—as a consequence of negligence or absence of safety protocols, either losing freedom and becoming a victim of coercion, abuse or threats trapped inside a modern-slavery network. These phenomena are linked as the kind of tasks that forced workers usually execute are mainly hazardous and risky jobs, particularly in agriculture and construction [21], and they provide a clear measure of the most basic component of the quality of employment—*safety and ethics of employment*—according to the seven-dimensions pyramidal model proposed by Körner, Puch [22] and developed by UN [23].

Second, decoupling analysis constitutes a method of determining if the ongoing economy dynamics drive the economy to the right sustainable development path. The concept of "decoupling" in a general sense refers to the phenomenon in which economic growth continues its course while the undesirable consequences and negative externalities triggered by it are reduced (i.e. environmental damages, use of resources and generation of waste, etc.). In the words of the OECD (the Organisation for Economic Co-operation and Development), the concept can be summarised as "breaking the link between *environmental bads* and *economic goods*" [24]. While decoupling analysis has blossomed in the environmental sphere (see for instance the recent examples of [25] for the Eropean Union and [26] for a comparison between United States and China), the social dimension has received less attention despite the negative impacts of indecent work are a current issue inside the international agenda. The decoupling analysis of the transport sector fatalities of [27] is one exception that proves the need for additional research. In this article, we focus on social impact decoupling understood as reducing the negative social impacts of any resources used or economic activities undertaken. We perform a worldwide analysis to determine which kind of decoupling -strong, weak or expansive/recessive, either positive or negative—in terms of social footprint predominates in 2000–2008 (expansion period) and in 2008–2013 (crisis period) according to the classification by Vehmas, Malaska [28]. The division of the time series into two periods is motivated by the general consensus about taking 2008 as the turning point in which the recession emerged and spread worldwide. Despite the timing of the economic crisis differs among countries, we consider that the comparison of these two stages can provide an interesting picture that captures possible changes in global trends triggered by the economic collapse in 2008.

Finally, it is common practice to examine the performance of the most developed economies in their efforts to delink negative externalities from production [29], but it is not as common to do so in terms of consumption. Footprint measures allow quantifying the consequences of consumption patterns instead of production patterns in order to determine if an economy's lifestyle is contributing to achieve an effective worldwide decoupling of negative impacts. We focus on decoupling trends from a consumption perspective in the 2000–2013 period in both the United States (USA) and the European Union (EU-28) comparing them with the production or domestic perspective using the same framework. Previous findings on the quality of labour caused by affluent regions' lifestyles suggest the different roles of high income and low-income regions [9]. For this reason, we find interesting to examine more in-depth the decoupling process from bad working conditions in two of the most affluent regions (EU-28 and USA). In this way, it can be assessed whether developed regions' declarations of intent concerning healthy and safe local production techniques are aligned with their efforts to improve international trade relationships. Furthermore, the use of footprint measures removes the illusion of decoupling or "virtual decoupling" that could result from looking only at domestic production, ignoring the lifestyle as driver of social distant damages and even the possibility of outsourcing production intensive in negative impacts to another nation or region in the world [30, 31]. In fact, authors like Wiedmann, Schandl [32] and Akizu-Gardoki, Bueno [33] point out that the analysis of decoupling between economic growth and material or energy impacts under footprint criterion reveals a poorer performance of developed nations than the one declared by those regions.

## Materials and methods

### Decoupling analysis

There are two main kinds of decoupling: resource decoupling, which refers to increasing resource productivity, i.e. requiring fewer inputs per unit of economic activity; and impact

decoupling, which is understood as reducing negative impacts per unit of economic activity [34]. In this work, we focus on social impact decoupling. The concept of either resource or impact decoupling is strongly related to the so-called intensity effect, which implies that individual economic sectors reduce their pressure intensity over time due to technological, social or environmental improvements. However, there are two additional effects leading to changes in direct pressures from production/consumption that might generate the illusion of decoupling: the production/consumption mix effect, in which changes in impact intensity or resource productivity are due to changes in the structure of the economy, and the output/demand effect, in which changes in total economic output or demand trigger changes in the resource or impact that is being assessed [35].

Furthermore, there are different classifications of decoupling. First, the European Commission [36] classifies it as either relative or absolute. On the one hand, relative decoupling of resources or impacts happens when the resources used or the negative impact analysed grow at a slower rate but positive than a relevant economic indicator [37]. On the other hand, absolute decoupling refers to the situation in which the use of resources or the undesired impact declines or remain stable independently from the pace followed by economic growth. While relative decoupling has been commonly observed in developed economies like EU in the last decades, absolute decoupling is an odd phenomenon that requires higher efforts to increase resource productivity and efficiency faster than the growth rate of the economy [34, 37, 38]. Second, following Vehmas, Malaska [28] classification and its application to transport made by Tapio [39], eight different situations can be distinguished as a function of the growth rate of the economic driver and the growth rate of the impact indicator of interest.

Using the concept of "social elasticity" developed as an extension of the ecological elasticity proposed by York, Rosa [40]–which is exposed in next subsection—the classification of Vehmas, Malaska [28] can be implemented according to the criteria exposed in Fig 1. There are three kinds of desirable decoupling: strong, in which the social negative impact falls in average terms over the period while Gross National Expenditure (GNE) increases ($\varepsilon^s_{period\ i} < 0$); weak, in which both the social and economic indicator grow in average terms, but the social impact does so in less than a proportional way ($0 < \varepsilon^s_{period\ i} < 0.8$); and recessive, in which both variables have a negative average growth, but the social impact declines more intensely than final demand ($\varepsilon^s_{period\ i} > 1.2$). On the other hand, there are three kinds of situations in which undesirable negative decoupling arises: strong negative decoupling, which corresponds to the situation in which the negative social impact grows in average terms over the period while GNE falls ($\varepsilon^s_{period\ i} < 0$); weak negative decoupling, which implies that both indicators exhibit a negative average growth rate over the period but the decline of GNE is more intense ($0 < \varepsilon^s_{period\ i} < 0.8$); and expansive negative decoupling, in which both indicators grow in average terms over the period, but the negative social impact does so more than proportionally ($\varepsilon^s_{period\ i} > 1.2$). Only in the case in which elasticity lies between 0.8 and 1.2 can it can be claimed to demonstrate coupling between the indicator and GNE.

## Social elasticity

The general concept of elasticity refers to the percentage of change in a dependent variable from a one per cent change in an independent variable that has a causal influence on the former with other factors held constant. Relying on this concept, environmental elasticity is defined as "the proportional change in environmental impacts due to a change in any driving force"[40]. In this work, we apply this idea to social impacts, defining social elasticity as the

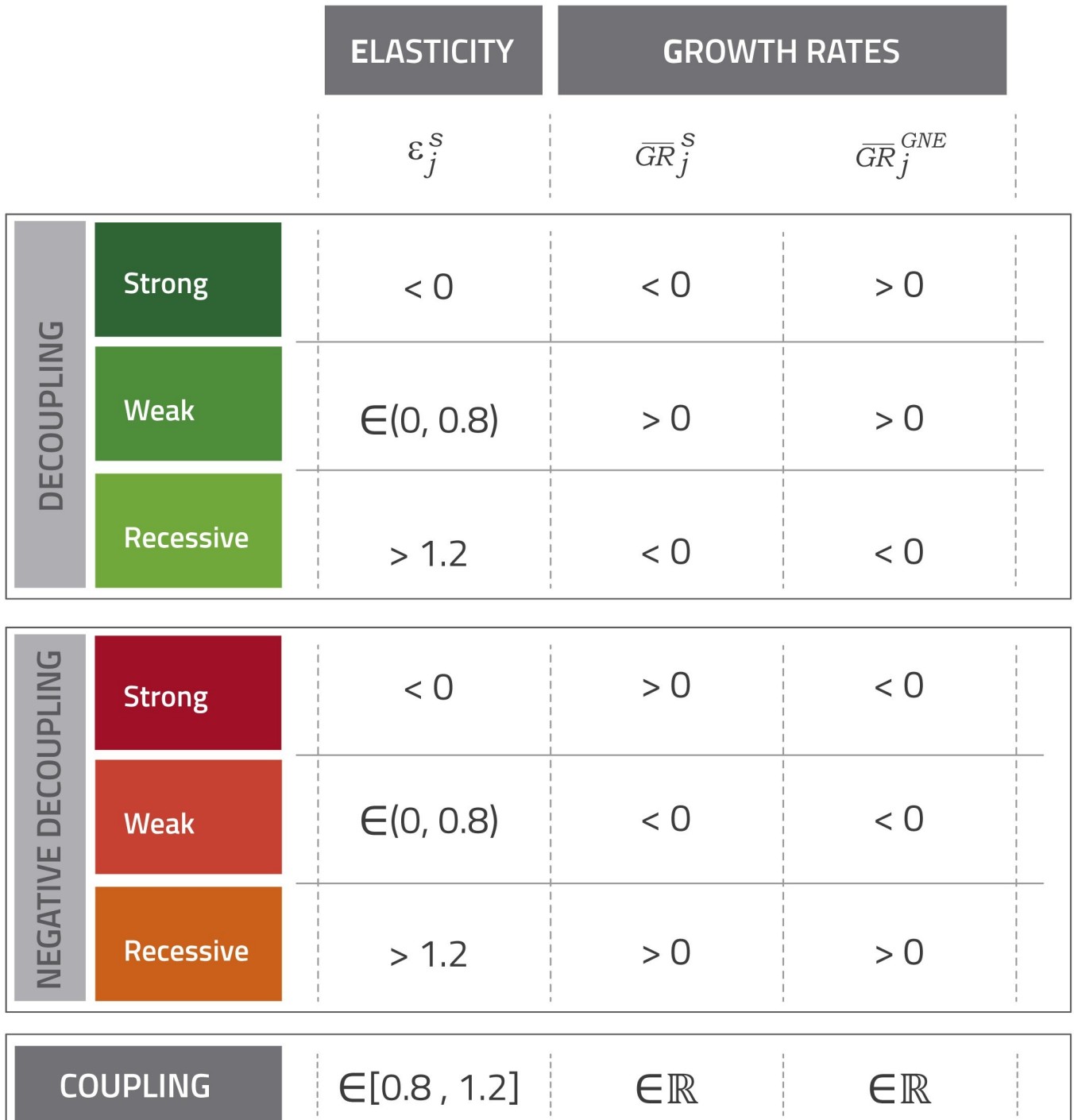

**Fig 1. Decoupling classification according to social elasticity.** Source. Own elaboration based on Vehmas, Malaska [28] and Tapio [39]. $\varepsilon_j^s$ represents the elasticity for social indicators with respect to GNE in each period j. $\overline{GR}_j^s$ stands for the average annual growth rate of each social indicator in period j and $\overline{GR}_j^{FD}$ denotes the average annual growth rate of GNE.

percentage of change in social impacts produced by a one per cent change in an economic driving force.

Social elasticity can be calculated for a single year as the ratio between the annual growth rate of the social impact (either in terms of footprint or in terms of producer responsibility) and the annual growth rate of the economic indicator chosen. In the case of a footprint perspective, expression [1] shows this, where $\varepsilon_t^s$ is the elasticity of the social indicator S in year t with respect to Gross National Expenditure (GNE), $GR_t^S$ stands for the annual growth rate of the social indicator S (in footprint terms) in year t, $GR_t^{GNE}$ stands for the annual growth rate of GNE in year t, $S_t$ represents the social indicator footprint value in year t and $GNE_t$ corresponds to GNE value in year t.

$$\varepsilon_t^s = \frac{GR_t^S}{GR_t^{GNE}} = \frac{\frac{S_t}{S_{t-1}} - 1}{\frac{GNE_t}{GNE_{t-1}} - 1} \tag{1}$$

Where $S \in$ {fatal injuries, non−fatal injuries, forced labour}; $t \in$ [2001,2013]

However, when working with periods of time spanning over several years, the calculation requires additional steps. First, we calculate the growth index ($GI_t^i$) for each of the variables $i$ (three social indicators and GNE) in each year $t$ in the time periods 2000–2013. Second, we calculate the geometric average of the growth index in each of the two periods selected (2000–2008 and 2008–2013) to obtain an average annual growth index ($\bar{GI}_j^i$) of each social indicator and GNE, where $j$ stands for the time period (2000–2008 or 2008–2013). Notice that working with growth indexes instead of growth rates avoids indeterminacy when applying the geometric average. Using the average annual growth index, the average annual growth rate ($\bar{GR}_{period\ i}^S$) can be obtained just by subtracting one unit. Finally, elasticity for each period is retrieved using a similar procedure to that shown in [1] just by substituting annual growth rates with the average annual growth rates.

In the case of a production perspective, the calculation method would be equivalent by introducing the social indicators in terms of PBA instead of CBA and by substituting GNE by Value Added (VA).

## The MRIO analysis

The standard extended multiregional input-output model (E-MRIO) is a well-established quantitative technique to measure the production requirements and the associated environmental and economic impacts, to meet a selected level of GNE across the whole supply chain [41–43]. Given increasing social concerns, an extension into the social dimension is the natural next step [44]. In the MRIO model framework, regions and countries are included with their own technology, and trade is divided into intermediate trade, with specific industry destinations, and final trade. The basic E-MRIO equation is as follows in expression [2]:

$$F = \hat{f}(I - A)^{-1}\hat{y} \tag{2}$$

where $\hat{f}$ is the target factor (either environmental or social) as a diagonalised vector per unit of output, $(I-A)^{-1}$ is the Leontief inverse matrix, and $\hat{y}$ is the diagonalised final demand per country [45]. $A$ is the matrix of technical coefficients in a MRIO context, providing a detailed sector-by-sector and region-by-region domestic intraregional structure and the trade matrixes from one region to another. The extension of the model to compute different social impacts and footprints is estimated by pre-multiplying the Leontief Inverse Matrix by target factors provided by different satellite accounts. The diagonalisation of those target factor vectors ($\hat{f}$) enables the estimation of multipliers and results, both in matrix form [46–48]. The resulting matrix $F$ can be analysed in different perspectives. By row, the $F$ matrix shows the distribution of impact that occurs in one sector of a country when produced to attend all sectors and

countries. This is the so-called production based-approach (PBA). Conversely, by column, the *F* matrix yields the impacts from across the world and across sectors required in the production of one unit of final demand in a country. This is the consumption-based approach (CBA) or factor footprint concept. We follow the CBA criteria to include all the global production chains in order to calculate some social footprints related to the quality of labour and job conditions, using data on non-fatal injuries, fatal injuries and forced labour by sector and country or region as factors. As a result, we evaluate the existence of decoupling by accounting not only for the quality of labour and job conditions at home (that are the only ones measured by PBA measures), but also for these circumstances abroad, along with the global production chain of goods and services imported to meet the countries' final demand. Current trends exhibit a general alignment of environmental footprint measures with Gross Domestic Product (GDP) per capita, even if the countries' profiles in PBA terms are different, which implies that decoupling negative impacts from consumption patterns can be harder to achieve than decoupling these impacts from national production [49].

### Data sources

The multi-regional database that the MRIO model relies on is the World Input–Output Database (WIOD) in its 2016 release [50, 51]. This source provides an annual series of multi-regional input-output tables from 2000 until 2014, built with data from diverse regions' national accounts and international trade statistics. The WIOD tables employed in the model have been previously deflated using the deflators provided by the same source and taking 2010 as the base year. The model considers 44 regions as given by the WIOD and gathers the original 56 sectors into 14. Regional and sectoral structures are detailed in Table A and Table B in S1 Appendix. In addition, the study of social decoupling assessed in this paper requires data on indecent labour—not only modern slavery, but also occupational injuries and fatalities—with a regional and sectoral disaggregation suitable for the calculation of footprints in a MRIO model, following the path settled by Gómez-Paredes, Yamasue [52] and Alsamawi, McBain [15]. Therefore, the WIOD database has been complemented with a satellite account composed by a set of self-compiled social indicators [53], whose generation process is described in S2 Appendix. All results and figures are based on these two datasets.

### Results

#### Worldwide social decoupling

The analysis of worldwide decoupling is implemented through the novel concept of social elasticity applied to the classification proposed by Vehmas, Malaska [28]. Our results show that most countries achieved the goal of social decoupling from fatal and non-fatal injuries at work, both in the expansion years and in the crisis period (see Fig 2), while the evolution of forced labour footprint decoupling is not favourable. The strongest case of decoupling is found for fatal injuries, as it shows the highest distance to the coupling zone, while the risk of no decoupling seems to be higher during the evolution of forced labour footprint. Looking at footprints for both kind of injuries, decoupling happens because the proportional change in both kinds of injuries is lower than the percentage of change in GNE. Besides, there is a global trend of moving from weak decoupling in the expansion period to recessive decoupling in the crisis one. This is shown in panels c and d in Fig 2 by the bullets cloud movement from blue (expansion period) to orange (crisis period). Decoupling trends of forced labour differ considerably from occupational injuries as panel b in Fig 2 shows: in the expansive period, weak positive decoupling predominated, but from 2008 onwards an involution seems to be happening, since the most common types are negative (both strong and weak) and the number of regions

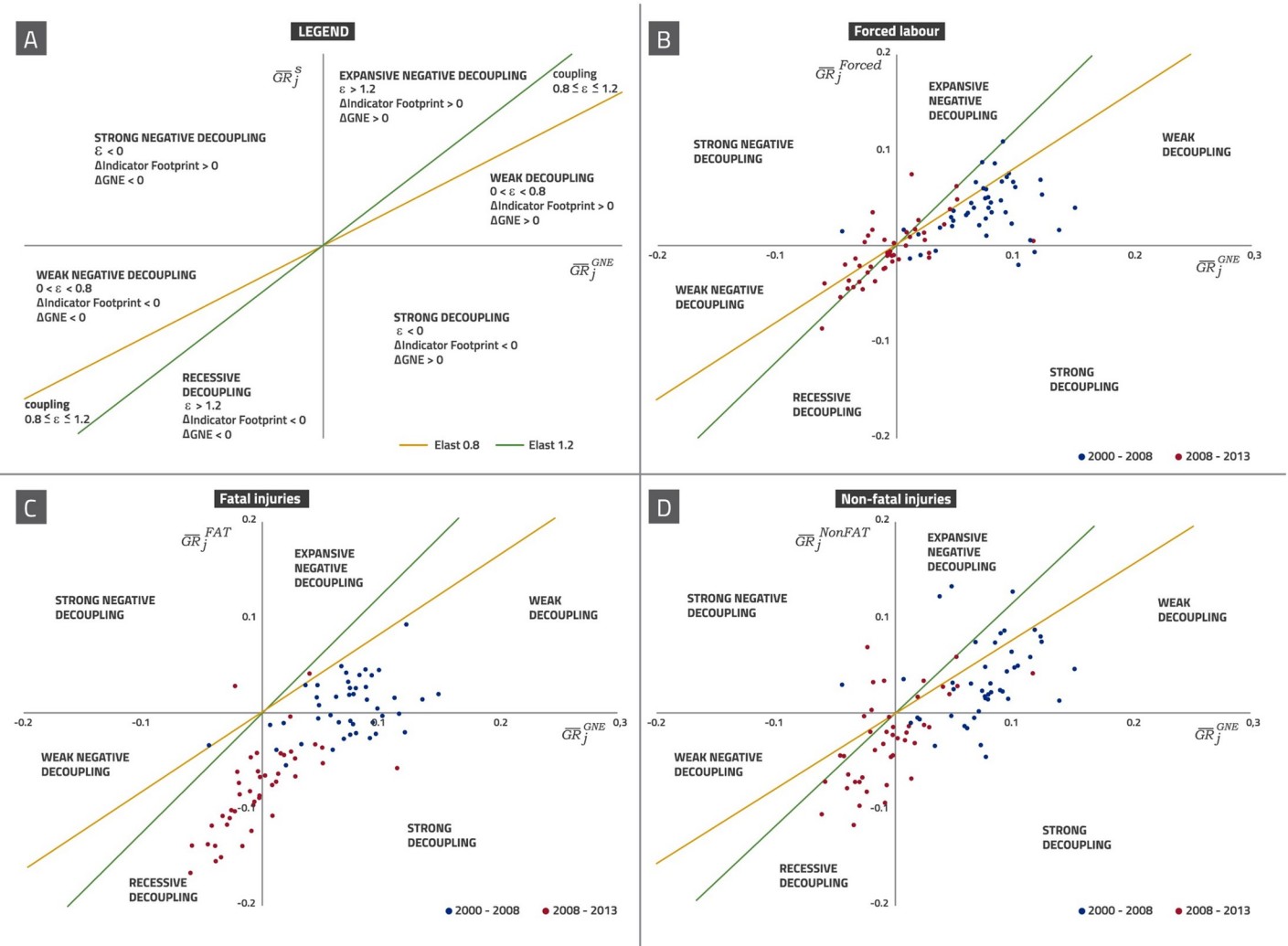

**Fig 2. Fatal injuries, non-fatal injuries and forced labour worldwide decoupling.** Worldwide data (44 regions). 2000–2008 and 2008–2013. Source: Own elaboration based on Timmer, Dietzenbacher [50] and Timmer, Los [51]. The vertical axis represents the average growth rate of the social impact of interest on each of the two periods plotted, while the horizontal axis represents the average growth rate of GNE on each of these two periods. The social indicators used in the calculus of elasticities are expressed in CBA terms. Panel a shows the legend as this classification is based on social elasticity concept with respect to real GNE according to Vehmas, Malaska [28] and Tapio [39] (for further detail on this decoupling classification, please check Fig 1 in "Decoupling analysis" section inside Methods). Panel b provides results for forced labour footprint, panel c for fatal injuries footprint and panel d for non-fatal injuries footprint. The specific behaviour of each country is provided in Table A in S3 Appendix.

experimenting no decoupling has also raised (see Table A in S3 Appendix for further country details). Despite GNE fell in this recessive period, forced labour footprints fell at a lower rate or even increased in most regions. The same analysis based on social elasticities calculated from the domestic indicators (PBA perspective) reveals stronger positive worldwide decoupling by the three indicators in the expansive period (see Figs A, B and C in S4 Appendix). In the second period, CBA analysis shows recessive decoupling as the predominant kind while PBA perspective leads to a wider variety of typologies. This is precisely a relevant result of this comparison among criteria: in the cases of fatal and non-fatal injuries, the production perspective leads to more dispersion among countries than the consumption perspective in the two periods analysed. The dispersion in PBA terms exposes the differences in direct incidences of social negative impacts among regions, which vanish when allocating these impacts according

to consumption patterns since regions generating high impacts domestically export them to developed countries. Forced labour constitutes a particular case in the recessive period: there is a worse overall performance in domestic terms than in the footprint ones, despite some developed regions that were in recessive decoupling under the CBA perspective show weak or even strong positive decoupling when switching to PBA (for instance, Belgium, Germany and France).

As can be found in Table A in S3 Appendix, it is remarkable that for all the indicators, there is a common trend shared by some EU countries that have moved from strong and weak decoupling in 2000–2008 to recessive decoupling positions in 2008–2013. Among these regions, we find those that suffered the worst consequences of the financial and economic crisis, such as Spain, Greece, Italy, Portugal [54] and other smaller economies within the EU, but also stronger economies like Belgium, Denmark or Germany. The cause behind this process is the effect of the economic recession, which motivated a higher fall in the three indicators' footprints than in the GNE. Focusing on the decrease of regions with positive types of forced labour decoupling (in 2000–2008, there were 7 out of 44 regions with no positive trends of decoupling, while in 2008–2013 there were 22), some European economies exhibit weak or even strong negative decoupling in the recessive period, while in the expansive one they didn't (for instance, Great Britain and Ireland in the first case and Austria, Russia and Netherlands in the second). Some other developed regions, like Australia or Norway, have moved from a positive decoupling process to a coupling trend, which shows a setback in social sustainability of consumption habits which will be analysed in more detail for EU in the next section. In addition, Mexico and India move to the worst kind of footprint decoupling classification (strong negative) in the second period. In this sense, although more research on the factors behind these social elasticity changes is needed, those cases can be seen as exceptions or as examples that suggest the fragility of decoupling, denoting that moving from decoupling to a situation of growing coupling is not only a possibility, but a reality.

## European Union and the United States: Social footprints and decoupling

The previous section shows the worldwide decoupling dynamics, but it lacks from looking detail at the factors that characterize the observed trends. To do so, we focus on the consumption patterns of the two major developed economies, the EU and the USA. These regions accounted for 47% of worldwide GNE in 2013 [51], so the evolution of their decoupling patterns between consumption and negative social impacts have an outstanding relevance inside global trends. The analysis of EU and USA labour footprints in terms of decoupling is implemented following OECD [38] guidelines. The preliminary graphical analysis of the footprint of each working conditions indicator and GNE trends in the two regions allows for some initial overview of the progress in decoupling both by footprint measures and by domestic production ones (Figs A and B in S5 Appendix). First, decoupling is more evident for the domestic production measures, displaying absolute decoupling in all cases, except in forced labour for USA that shows relative decoupling. Second, focusing on the footprint indicators, on the one hand, the expansive period 2000–2008 led to relative decoupling of bad working conditions in both regions, since the three indicators footprints grew though less rapidly than GNE. However, in USA theres is a change towards absolute decoupling that began in 2006, probably linked to the earliest crisis beginnings. On the other hand, while USA maintained these trends in the recessive period of 2008–2013, the EU showed recessive decoupling in fatalities and unclear results for the rest of the social indicators that might be pointing towards a modest recessive decoupling process, according to the classification of Vehmas, Malaska [28].

As the OECD [38] suggests, the ratio between each social impact variable footprint and GNE allows to look for decoupling and, at the same time, our method allows us to zoom in on the details in the imported part of each footprint. A decline in this indicator over time can be taken as evidence of decoupling. The most interesting feature of our proposal is that it allows for decomposing the imported part of each footprint according to the country of origin, which might be useful in determining possible sources of decoupling or specific areas in which no decoupling takes place. This kind of analysis can be very relevant for policymakers since it rules out the output/demand effect and, with the proper disaggregation of results, allows to extract conclusions regarding the production mix effect and intensity effect behind the trends. Fig 3 shows each region of origin's contribution to each social footprint—relativized according to real GNE to obtain a comparable intensity measure of decoupling—throughout the 2000–2013 period.

The first aspect to highlight is the relevant share of imports in the footprint of each indicator of working conditions. Over 80% of the total forced labour linked to final demand in EU and USA happened outside their borders due to the virtual bad labour imports coming mainly from developing countries and regions such as China, RoW or India. The import shares are slightly lower for the other two social indicators, but still account for more than half of the social footprints in most cases: around 44% (USA) and 70% (EU) for fatal injuries and around 69% (USA) and 55% (EU) for non-fatal injuries in 2013. These high shares of social impacts embodied in imports confirm that the analysis of bad labour conditions of domestic production, excluding international trade and imports of intermediate inputs and final products, provide underestimated results.

For the three impacts considered, the whole footprint and its domestic and imported parts, both in EU and USA, have been reduced over time, more intensely in the case of EU's non-fatal injuries footprint and USA's fatal injuries footprint and more modestly in the case of forced labour. These trends confirm the decoupling process found at a global level for both injuries indicators and the limitations of the path followed by forced labour footprints. They indicate that, in the cases of EU and USA, decoupling of occupational injuries is shared by both production at home and abroad to meet the needs of domestic demand, while, in the case of forced labour, the imported part of both regions slightly declines. The high participation of Eastern European countries in the EU's domestic part of both fatal injuries and forced labour footprints is an inauspicious element.

It is remarkable that the intensity of the social footprint in relation to the GNE is similar for USA and EU, despite the differences in their economies. The exception is the case of non-fatal injuries, in which the intensity is higher for EU due to high domestic values, mainly in the first years of the period 2000–2008. Injuries at work constitute a special case in which, for both EU and USA in the case of fatal and non-fatal injuries, the participation of the domestic part is as high or even higher than the participation of the imported part in the whole footprint. This phenomenon can be due to three facts: first, both areas present a strong and developed social and labour protection performance, so most workers are accounted for in the social security system and are able to report their work accidents and injuries. Secondly, these regions accept a wide spectrum of cases as occupational accidents that can lead to time off work, including those related to psychological risks. According to an EU OSHA report [55], 20% of workers in EU-15 and 30% of workers in EU-10 claimed to suffer a stress-related health problem in 2009. The Matrix Insight [56] report under the EU 2008–2013 health programme states that mental illness represents a significant percentage of the overall health conditions affecting European workers. In contrast to areas such as India, China or Indonesia, developed regions are highly aware of these types of conditions that can be disabling elements at work, and report them in their statistics. Thirdly, according to the economic structure provided by the WIOD input-

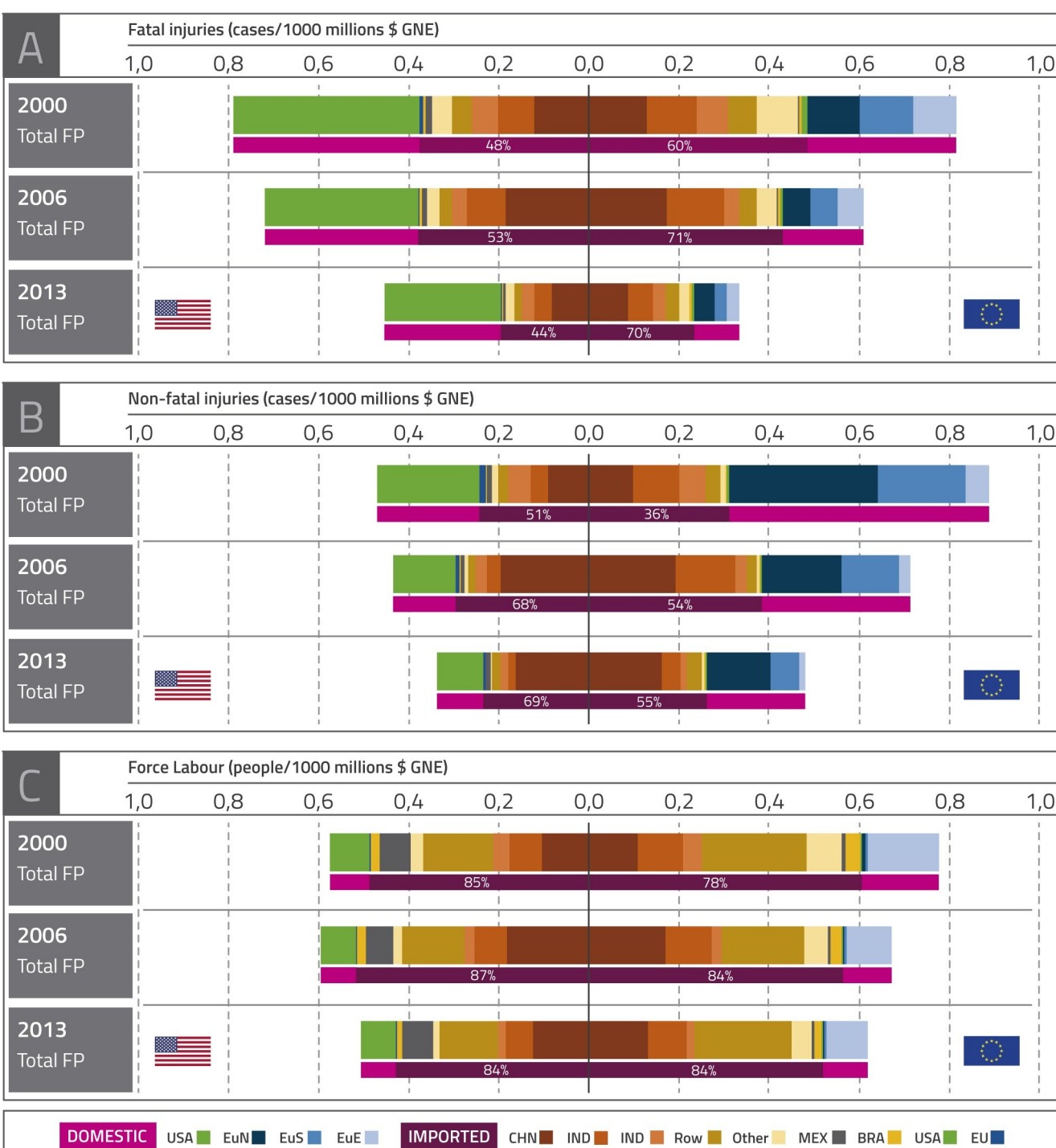

**Fig 3. Fatal injuries, non-fatal injuries and forced labour footprints for EU-28 and USA.** Relativized using real GNE. 2000–2013. Source: Own elaboration based in Timmer, Dietzenbacher [50] and Timmer, Los [51]. Fig 3 shows the fatal injuries (panel a), non-fatal injuries (panel b) and forced labour (panel c) footprints of EU and USA. These footprints appear divided into their domestic and imported part (upper bar for each year), and the imported part appears also split by origin country (lower bar for each year). In addition, the domestic part of UE's footprints are disaggregated into three sub-regions: North of EU (denoted as EuN, including Austria, Belgium, Germany, Denmark, Finland, France, United Kingdom, Luxembourg, Netherlands and Sweden), South of EU (denoted as EuS, including Cyprus, Greece, Ireland, Italy, Malta and Portugal) and East of EU (denoted as EuE, including Bulgaria, Czech Republic, Estonia, Croatia, Hungary, Lithuania, Latvia, Poland, Romania, Slovak Republic and Slovenia). However, when analysing the imported part of USA's footprints, EU is plotted as a unique region with no sub-regions (EU) to provide clearer results. The region "Others" include Australia, Canada, Switzerland, Japan, Korea, Norway, Russia, Turkey and Taiwan.

output tables, the expenditure of industrial sectors in the private health sector per worker is much higher in developed economies like USA or EU. Assuming that the greater expenditure per worker, the more effective diagnosis of injuries and illnesses, higher incidences of occupational injuries can be expected in those countries in which the aforementioned ratio is greater.

The analysis of forced labour is also outstanding. Despite a reduction in the intensity, both for EU and USA, the domestic and imported footprint intensity remains fairly stable for the whole period. In the year 2000, 85% of forced labour was imported in USA, with 84% in 2013. The case of EU is similar but presents some interesting differences. The figures are higher than those for USA for the whole period. Imported participation increases during the whole period, from 78% to 84%. The differences regarding the quality of labour within EU as shown in [57, 58] are identified here again, in this case through the forced labour indicator. Eastern EU countries account for almost all the domestic contribution to footprint intensity. It might be due to certain self-reinforcing factors such as the establishment of refugee corridors along Eastern Europe, the liberalisation of borders and the institutional weakness of some transitional economies, which allow migrants to be prone to exploitation and human trafficking intensely in this area [14, 59].

The analysis of the imported part shows that China, India and Indonesia are the main suppliers of the fatal and non-fatal injuries footprint for both EU and USA as destination regions. Emerging countries from Africa, Latin America and Asia included in the RoW region and other countries (mainly Russia, Turkey and Taiwan) are also relevant when talking about a fatal injuries' footprint. The participation of each region in the total amount of the footprint is stable for the whole period except for China, which increases its share. For USA, additional actors appear in comparison with EU: Mexico in the case of both kinds of injuries and EU in the first years of the series for non-fatal injuries. Concerning forced labour, Mexico emerges as a remarkable supplier of USA while Brazil arises as one relevant supplier for both USA and EU. The RoW is the most influential supplier in both cases since most of this kind of work, worldwide, proceeds from the primary sector in Africa and Latin America [12, 60].

The observed trend of increasing shares of imports in the three footprint intensities in the expansion years (up to 2006 as shown in Fig 3) along with an opposite trend in the crisis (up to 2013, the last year available) draw attention to the risk of a decoupling progress in the recovery years with decreasing footprints, but with higher imported shares, like in the expansion period prior to the crisis. This risk in the recovery would indicate a faster improvement of working conditions at home than abroad, where paradoxically, there is more room for improvement, and that affluent regions would be more conscious of local damage than of distant damage.

## Discussion

In a globalised world, the path to sustainable development requires efforts that equally take not only conditions at home but also abroad into account. This paper assesses how global production chains and consumption patterns and lifestyle affect social impact decoupling, first at a worldwide level and later focusing on two rich regions: EU and USA.

Our results show that most countries achieved the goal of improving working conditions embodied in their societies' consumption in the 2000–2013 period. The decoupling from bad working conditions revealed by occupational injuries, fatal and non-fatal, and forced labour embodied in the countries' final demand was even stronger in the crisis years. Although this is a positive result, it points to worse social footprint behaviour during expansion, so close attention must be paid to prevent economic recovery from reversing the decoupling trends observed. In addition, our results show that domestic decoupling seems to be stronger than footprint decoupling in the expansive period 2000–2008, while in the recessive period a

production perspective provides a wider variety of typologies, revealing the differences among countries in terms of incidence of negative social impacts.

At a worldwide level, a decoupling process turns out to be more solid in the case of the fatal injuries footprint, while the risk of coupling is higher in the case of the forced labour footprint. Despite having a random component, fatalities at work have shown a certain correlation with the degree of institutional progress achieved [21] [61], so it is reasonable to expect the corresponding footprint to diminish over time since developing countries will become more conscious of the importance of labour rights and the need to implement occupational health and safety measures. On the contrary, forced labour is a hidden phenomenon strongly linked to social relationships, such as dependence of isolated communities, the vulnerability of certain minorities and the absence of choice for people trapped in clandestine exploitation networks [12]. Therefore, lowering the figures requires complex and non-standardised actions that can be out of reach for non-mature institutions in developing countries.

While environmental impacts such as emissions have a deterministic nature linked to the magnitude of production, use of energy and the level of technological progress, safety-at-work indicators include a random component that may hinder the decoupling analysis. Firstly, in the environmental sphere, developing countries by adopting techniques from developed countries usually tend to achieve a reduction in the pollution intensity [62]. However, in safety-at-work terms, adopting more advanced production techniques can either reduce incidents if the new processes are safer or increase them if implemented too quickly without appropriate training and prevention protocols [21] or when better labour protection and rights lead to increased coverage and reporting of incidents by workers. Even though, in some sectors environmental decoupling (decarbonisation) can be more difficult to achieve than the social one (fatalities), as in the transport sector [27]. Secondly, the random counterpart in this kind of social indicators suggests that there might be a certain intrinsic limit in the decrease of occupational injuries. In the case of fatalities and accidents at work, figures could tend to stabilise when the risk levels achieved are taken as acceptable [63]. An interesting extension in future research would be to estimate this lower bound, taking as a proxy for each indicator the conditions achieved in the country with the best performance. Therefore, considering these two divergence sources between environmental and occupational-health impacts, the nature of safety-at work figures might blur the analysis of causes and effects concerning decoupling trends. Measures introducing social certification and standards like the environmental ones [64] can help in achieveing social upgrading, although some case studies provide mixed results like the improvement of environmental performance but not better social conduct than non-certified firms [65].However, in contrast with occupational injuries, forced labour is not inherently dependent on the production, which exhibits the possibility of "producing better" in slavery terms as a reality and reveals the power of change that production and consumption decisions have to achieve full decoupling of forced labour in the future.

## Supporting information

**S1 Appendix. Regional and sectorial structure.**
(DOCX)

**S2 Appendix. Social impacts database.**
(DOCX)

**S3 Appendix. Worldwide decoupling (footprint perspective): Detailed results.**
(DOCX)

**S4 Appendix. Worldwide decoupling analysis based on PBA measures.**
(DOCX)

**S5 Appendix. Preliminary decoupling analysis.**
(DOCX)

## Author Contributions

**Conceptualization:** Ángela García-Alaminos, Fabio Monsalve, Jorge Zafrilla, Maria-Angeles Cadarso.

**Data curation:** Ángela García-Alaminos.

**Formal analysis:** Ángela García-Alaminos, Fabio Monsalve, Jorge Zafrilla, Maria-Angeles Cadarso.

**Software:** Fabio Monsalve.

**Writing – original draft:** Ángela García-Alaminos, Fabio Monsalve, Jorge Zafrilla, Maria-Angeles Cadarso.

**Writing – review & editing:** Ángela García-Alaminos, Fabio Monsalve, Jorge Zafrilla, Maria-Angeles Cadarso.

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
