## [Decision Letter · Decision Letter 0]

14 Nov 2019

PONE-D-19-26281

Unmasking social distant damage of developed regions’ lifestyle: A decoupling analysis

PLOS ONE

Dear Ms. García-Alaminos,

Thank you for submitting your manuscript to PLOS ONE. After careful consideration, we feel that it has merit but does not fully meet PLOS ONE’s publication criteria as it currently stands. Therefore, we invite you to submit a revised version of the manuscript that addresses the points raised during the review process.

Both reviewers find merit in your manuscript, however, they have some raised some issues that should be addressed. Critically, you should address Reviewer #2 point regarding direct and indirect (footprint) decoupling. I think that the manuscript would increase in quality if you analyze the direct and indirect decoupling in the same framework. Please also address all other minor points raised by both reviewers.

We would appreciate receiving your revised manuscript by Dec 29 2019 11:59PM. To enhance the reproducibility of your results, we recommend that if applicable you deposit your laboratory protocols in protocols.io, where a protocol can be assigned its own identifier (DOI) such that it can be cited independently in the future. For instructions see: http://journals.plos.org/plosone/s/submission-guidelines#loc-laboratory-protocols

We look forward to receiving your revised manuscript.

Kind regards,

Jordi Paniagua

Academic Editor

PLOS ONE

Journal Requirements:

2. Please do not include funding sources in the Acknowledgments or anywhere else in the manuscript file. Funding information should only be entered in the financial disclosure section of the submission system. https://journals.plos.org/plosone/s/submission-guidelines#loc-acknowledgments

* Thank you for stating the following in the Acknowledgments Section of your manuscript:

"We thank the Ministry of Economy and Competitiveness of

Spain for funding the research project “ECO2012-33341 that led to this paper. Ángela

García-Alaminos also thanks the European Social Fund and University of Castilla-La

Mancha support through the Regional FPI program (PRE-18: Contratos predoctorales

para la formación de personal investigador en el marco del plan propio de I+D+i,

susceptible de cofinanciación por el Fondo Social Europeo).".

 "F.M., J.Z. and M.-A.C thank the Ministry of Economy and Competitiveness of Spain for funding the research project “ECO2012-33341 that led to this paper. http://www.mineco.gob.es/

Á. G-A. also thanks the European Social Fund and University of Castilla-La Mancha support through the Regional FPI program (PRE-18: Contratos predoctorales para la formación de personal investigador en el marco del plan propio de I+D+i, susceptible de cofinanciación por el Fondo Social Europeo).

https://ec.europa.eu/esf/home.jsp

https://www.uclm.es/

The funders had no role in study design, data collection and analysis, decision to publish, or preparation of the manuscript.".

Reviewers' comments:

Reviewer's Responses to Questions

**Comments to the Author**

1. Is the manuscript technically sound, and do the data support the conclusions?

Reviewer #1: Yes

Reviewer #2: Partly

2. Has the statistical analysis been performed appropriately and rigorously? 

Reviewer #1: Yes

Reviewer #2: N/A

3. Have the authors made all data underlying the findings in their manuscript fully available?

Reviewer #1: Yes

Reviewer #2: No

4. Is the manuscript presented in an intelligible fashion and written in standard English?

Reviewer #1: Yes

Reviewer #2: Yes

5. Review Comments to the Author

Reviewer #1: I consider that authors should take into account the following:

1. In the last paragraph of the Introduction it is said that: "Furthermore, it removes the illusion of decoupling that could result from looking only TAT domestic production,....". What it is "tat"?

2. In the paragraph before "Insert fig 1 here", GNE is mentioned. But it is not defined until the second paragraph in the section 2.2.

3. When the social elasticity is defined, it is said that the geometric average of the growth index is calculated. When authors calculate the geometric average they understand that the growth is cumulative and it is not summative. This is similar to the average interest rate in compound capitalization. However, authors write "This method provides more accurate results than calculating a cumulative annual growth rate, which would assume growth every year in an unrealistic way". There is a mistake in the sentence. The geometric average implies a cumulative growth, while the arithmetic average implies a summative growth.

4. In the last paragraph of section 2.2 authors mention GDP, but it is not defined. I understand that it is Gross Domestic Product.

Reviewer #2: García-Alaminos et al.

This paper examined the decoupling between consumption and social labor problems (fatal injuries, non-fatal injuries, and forced labor). Overall, their technical quality is great and the authors test their research question using appropriate datasets. I still have several concerns about their analytical method, data-making, and open-data policy. If the authors address my following concerns appropriately, I am happy to recommend the editor to accept this paper.

Major comments:

- Section 2.1: There are several papers examine the resource decoupling using footprint accounting. For example, Weidmann et al. (2015) show many countries fail to achieve resource decoupling in footprint perspective even though many studies show resource decoupling in DMC. I recommend the authors to analyze the direct and indirect decoupling in the same framework. Currently, I am not convinced why the authors choose this type of decoupling analytical method. Several studies have already compared direct and indirect (footprint) decoupling and the authors justify to not follow the analytical framework (Weidmann et al. 2015).

- Figures: The quality of the figures is too low and I cannot interpret the figures correctly. Maybe this is not the authors' problem.

- SI P.8: I am not sure whether I understand correctly, but the sector disaggregation based on the share of low/high skilled workers is not an appropriate way to do this analysis. Please reconsider the method or simply drop forced labor analysis from this paper.

- Finally, the authors should open direct fatal and non-fatal injuries and forced labor datasets to prepare f vector in SI.

Wiedmann, T., Schandl, H., Lenzen, M., Moran, D.D., Suh, S., West, J., and Kanemoto, K. (2015). The material footprint of nations. Proc. Natl. Acad. Sci. U. S. A. 112, 6271–6276.

6. PLOS authors have the option to publish the peer review history of their article (what does this mean?). If published, this will include your full peer review and any attached files.

Reviewer #1: No

Reviewer #2: No

---

## [Author Response · Author response to Decision Letter 0]

10 Dec 2019

We would like to thank the very useful comments and suggestions of the two reviewers. We think they have improved the paper. In the following lines, we explain the changes introduced according to their suggestions (in blue) and some specific new in-text captions (in green).

Reviewer #1:

1. In the last paragraph of the Introduction it is said that: "Furthermore, it removes the illusion of decoupling that could result from looking only TAT domestic production,....". What it is "tat"?

It was a misprint. It should be “at” not “tat”. We have corrected it in the text.

2. In the paragraph before "Insert fig 1 here", GNE is mentioned. But it is not defined until the second paragraph in the section 2.2.

We have defined it the first time that appears. It is the Gross National Expenditure (GNE).

3. When the social elasticity is defined, it is said that the geometric average of the growth index is calculated. When authors calculate the geometric average they understand that the growth is cumulative and it is not summative. This is similar to the average interest rate in compound capitalization. However, authors write "This method provides more accurate results than calculating a cumulative annual growth rate, which would assume growth every year in an unrealistic way". There is a mistake in the sentence. The geometric average implies a cumulative growth, while the arithmetic average implies a summative growth.

Yes, we agree, it is a mistake indeed. We have corrected it by deleting the confusing sentence since it does not suppose a contribution. We don’t find necessary to explain why geometric mean is the most suitable one in the case of growth rates and growth indexes as it is a well-stablished practice in literature (see for instance Diakoulaki and Mandaraka (2007), which use the geometric mean annual rate of change of environmental impacts in a decoupling context too) .

4. In the last paragraph of section 2.2 authors mention GDP, but it is not defined. I understand that it is Gross Domestic Product.

We have defined Gross Domestic Product (GDP) at the end of the section 2.3.

Reviewer #2:

- Section 2.1: There are several papers examine the resource decoupling using footprint accounting. For example, Weidmann et al. (2015) show many countries fail to achieve resource decoupling in footprint perspective even though many studies show resource decoupling in DMC. I recommend the authors to analyze the direct and indirect decoupling in the same framework. Currently, I am not convinced why the authors choose this type of decoupling analytical method. Several studies have already compared direct and indirect (footprint) decoupling and the authors justify to not follow the analytical framework (Weidmann et al. 2015).

We agree that circumscribing the analysis to footprint criteria could not be enough to show its relevance, which precisely arises when these results are compared to production measurements. For this reason, we have introduced direct and indirect decoupling in the same framework following your recommendations. Our analysis still gravitates around a consumption perspective as we feel that it is one of our main contributions (we don’t have about other paper evaluating social decoupling from this perspective), but we think that the new strokes comparing these results with production measurements enrich and give a deeper perspective about the actual decoupling performance of regions and about its determinants. Let us clarify the details of this new outlook:

First, in the last paragraph of the introduction, we have enriched references supporting to the need of footprint measures in a decoupling context to justify our decision of paying special attention to this criteria. We have cited the concept of “virtual decoupling” proposed by Moreau and Vuille (2018) and we refer to Wiedmann et al. (2015) and Akizu-Gardoki et al. (2018) as previous works employing footprint measurements in a decoupling context to determine if developed nations are decoupling at the same extent as it seems when looking just at domestic production. 

Second, section 3.1. now compares worldwide footprint results with those referred as “domestic decoupling” along the text, which rely on the producer-based account criteria (PBA). Figures related to PBA assessment are shown in SI Appendix 3 in order to maintain coherence in the main text and not blurring too much the main line of argument of the paper. This comparison truly improves our conclusions as it shows that production perspective leads to more dispersion among countries in their decoupling trends. More specifically, the novel ideas introduced in the main text are the following ones:

The same analysis based on social elasticities calculated from the domestic indicators (PBA perspective) reveals stronger positive worldwide decoupling by the three indicators in the first period (see Figures SI3, SI4 and SI5). In the second period, CBA analysis showed recessive decoupling as the predominant type while PBA perspective leads to a wider variety of typologies. This is precisely a relevant result of this comparison among criteria: in the cases of fatal and non-fatal injuries, production perspective leads to more dispersion among countries than consumption perspective in the two periods analysed. The dispersion in PBA terms exposes the differences in direct incidences of social negative impacts among regions, which vanish when allocating these impacts according to consumption patterns since regions generating high impacts domestically export them to developed countries. Forced labour constitutes a particular case in the second period: there is a worse overall performance in domestic terms than in footprint measurement, despite some developed regions that were in recessive decoupling under the CBA perspective show weak or even strong positive decoupling when switching to PBA (for instance, Belgium, Germany and France).

Third, the preliminary decoupling analysis commented at the beginning of section 3.2. has been completed by introducing the PBA perspective too. Figures concerning domestic decoupling are shown in SI Appendix 4 so as not to get out of our main storyline path. In order to have a clearer comparison between the former figure concerning CBA (Figures SI.6) and the new one regarding PBA (Figures SI.7), we have expressed as indexes all of the variables plotted as it shows directly relative changes along time. We have introduced comments both in the SI Appendix 4 and in the manuscript. New ideas in the main text are the following: 

The preliminary graphical analysis of the footprint of each working conditions indicator and GNE trends in the two regions allows for some initial overview of the progress in decoupling both by footprint measures and by domestic production ones (Figures SI6 and SI7). First, decoupling is more evident for the domestic production measures, displaying absolute decoupling in all cases, except in forced labour for USA that shows relative decoupling.

Fourth, the split between direct (generated inside the borders of the region to cover its own demand) and indirect (imported to cover the demand of the region) impacts when evaluating the social footprint/GNE ratio of EU and USA along time (Figure 3 in the main text) was already exposed in the first version of the manuscript. However, we have modify Figure 3 to avoid redundant information and to make the analysis clearer at a glance. The first version of the main text contained comments about these two components of the footprint that are still on the manuscript.

Finally, the discussion section has been modified to introduce the highlights of the new comparison between footprint and domestic perspectives. 

- Figures: The quality of the figures is too low and I cannot interpret the figures correctly. Maybe this is not the authors' problem.

 We uploaded each figure in separate files with the highest resolution possible. Maybe, previous problems with the visualization happened when the PDF was generated automatically by the submission system, as the quality of the figures were low there. We don’t know if you could open those individual files. Nevertheless, we have redesigned each of the figures and tables hiring the services of an expert designer to improve the readability and the visualization of each figure in the main text (in all the cases the information showed is exactly the same as in the previous submission). Please, check the new figures provided in the resubmission.

- SI P.8: I am not sure whether I understand correctly, but the sector disaggregation based on the share of low/high skilled workers is not an appropriate way to do this analysis. Please reconsider the method or simply drop forced labour analysis from this paper.

 We appreciate your concern about this allocation method. When we faced this obstacle while developing the database we inquired about ways to deal with it. After a deep literature review, we found that this sector disaggregation method could be the most adequate one for two main reasons.

First, forced labour is characterized by several international organisms as a phenomenon happening mostly in low-skill activities such as mining, agricultural o manufacturing sectors (Fletcher, Bales, & Stover, 2005; ILO, 2012a; Srivastava, 2005; USBILA, 2016), being agriculture most probably the single largest sector in which forced labour happens (Belser, 2005). This lack of skills and formation is precisely one of the determinants that most commonly generate vulnerability among workers and that makes them prone to be trapped into networks operating with modern slavery (ILO, 2014). Low-skilled migrant workers, children and indigenous peoples are have among the most vulnerable collectives (ILO, 2012b), and all of them could be considered to be low-skill workforce (children and indigenous people being forced workers commonly haven’t had proper chances to acquire medium or high skills). This pattern applies not only to developing regions: research results confirm that forced labour in the UK is located in sectors characterized by low-skilled and low-paid labour (Geddes et al., 2013). Therefore, since we found enough references linking low-skilled activities and forced labour, we reckon that providing a sectoral disaggregation based on the shares of low-skilled workers among industries is a sensible practice.

Second, this disaggregation method has been already implemented by other authors when sectoral breakdown is required for indecent labour variables. Simas, Golsteijn, Huijbregts, Wood, and Hertwich (2014) generate forced labour and child labour indicators in a similar way as we do: they split original ILO data into 163 economic sectors by using each sector’s low-skilled labour share with respect to total low-skilled labour in the region as it is can be checked in the supporting materials of their aforementioned work. 

We are aware that it can constitute a limitation of the forced labour indicator, but this estimation was unavoidable in order to conciliate underlying datasets and our methodological approach based on a MRIO model. As state by Karstensen (2018), renouncing to the methods required to combine different datasets would jeopardize the scope of our analysis and leave behind potential conclusions that may be interesting and robust. It must be noticed that the sectoral allocation is used as an input of the model, but it is not exploited in our results analysis. We focus on global trends looking at a region-level instead of exploring industry-level implications, so the robustness of our conclusions doesn't depend on the industry allocation method.

In order to reinforce the explanation given in SI materials concerning this issue, we have extended it by introducing the previous justification behind this sectoral allocation method (see S2 Appendix).

- Finally, the authors should open direct fatal and non-fatal injuries and forced labor datasets to prepare f vector in SI.

All the datasets developed for this paper’s purposes have been uploaded and openly shared to Mendeley Data as stated in the Data Sources sections of the main manuscript and in the SI Appendix 2. They can be downloaded freely and used under license CC BY 4.0 by citing García Alaminos, Ángela (2019), “Social Indicators of Working Conditions Database”, Mendeley Data, v1. http://dx.doi.org/10.17632/6h5msdfjk2.1. Mendeley Data accomplishes all repository criteria recommended by PLOS to follow FAIR principles on data openness.

References

Akizu-Gardoki, O., Bueno, G., Wiedmann, T., Lopez-Guede, J. M., Arto, I., Hernandez, P., & Moran, D. (2018). Decoupling between human development and energy consumption within footprint accounts. Journal of Cleaner Production, 202, 1145-1157. doi:https://doi.org/10.1016/j.jclepro.2018.08.235

Belser, P. (2005). Forced Labour and Human Trafficking: Estimating the Profits. SSRN Electronic Journal. doi:10.2139/ssrn.1838403

Diakoulaki, D., & Mandaraka, M. (2007). Decomposition analysis for assessing the progress in decoupling industrial growth from CO2 emissions in the EU manufacturing sector. Energy Economics, 29(4), 636-664. doi:https://doi.org/10.1016/j.eneco.2007.01.005

Fletcher, L. E., Bales, K., & Stover, E. (2005). Hidden slaves: Forced labor in the United States. Berkeley Journal of International Law, 23(1), 47-96. 

Geddes, A., Craig, G., Scott, S., Ackers, L., Robinson, O., & Scullion, D. (2013). Forced labour in the UK. Retrieved from Bristol: https://www.basw.co.uk/system/files/resources/basw_120247-5_0.pdf

ILO. (2012a). Global Estimate of Forced Labour. Retrieved from Geneva: 

ILO. (2012b). Stopping forced labour and slavery-like practices - The ILO strategy. In. Geneva, Switzerland.

ILO. (2014). Strengthening action to end forced labour. Paper presented at the International Labour Conference 103rd Session 2014, Geneva, Switzerland. https://www.ilo.org/ilc/ILCSessions/previous-sessions/103/reports/reports-to-the-conference/WCMS_239814/lang--en/index.htm

Karstensen, J., Peters, G. P., & Andrew, R. M. . (2018). Trends of the EU’s territorial and consumption-based emissions from 1990 to 2016. Climatic Change, 151(2), 131-142. 

Moreau, V., & Vuille, F. (2018). Decoupling energy use and economic growth: Counter evidence from structural effects and embodied energy in trade. Applied Energy, 215, 54-62. doi:https://doi.org/10.1016/j.apenergy.2018.01.044

Simas, M., Golsteijn, L., Huijbregts, M., Wood, R., & Hertwich, E. (2014). The “Bad Labor” Footprint: Quantifying the Social Impacts of Globalization. Sustainability, 6(11), 7514. Retrieved from http://www.mdpi.com/2071-1050/6/11/7514

Srivastava, R. S. (2005). Bonded Labor in India: Its Incidence and Pattern. Retrieved from Geneva: 

USBILA. (2016). List of Goods Produced by Child Labor or Forced Labor. Retrieved from Washington, D.C.: 

Wiedmann, T. O., Schandl, H., Lenzen, M., Moran, D., Suh, S., West, J., & Kanemoto, K. (2015). The material footprint of nations. Proceedings of the National Academy of Sciences, 112(20), 6271-6276. doi:10.1073/pnas.1220362110

---

## [Decision Letter · Decision Letter 1]

22 Jan 2020

Unmasking social distant damage of developed regions’ lifestyle: A decoupling analysis of the indecent labour footprint

PONE-D-19-26281R1

Dear Dr. García-Alaminos,

We are pleased to inform you that your manuscript has been judged scientifically suitable for publication and will be formally accepted for publication once it complies with all outstanding technical requirements.

Well done!

With kind regards,

Jordi Paniagua

Academic Editor

PLOS ONE

Additional Editor Comments (optional):

Reviewers' comments:

Reviewer's Responses to Questions

**Comments to the Author**

1. If the authors have adequately addressed your comments raised in a previous round of review and you feel that this manuscript is now acceptable for publication, you may indicate that here to bypass the “Comments to the Author” section, enter your conflict of interest statement in the “Confidential to Editor” section, and submit your "Accept" recommendation.

Reviewer #1: All comments have been addressed

Reviewer #2: All comments have been addressed

2. Is the manuscript technically sound, and do the data support the conclusions?

Reviewer #1: Yes

Reviewer #2: Yes

3. Has the statistical analysis been performed appropriately and rigorously? 

Reviewer #1: Yes

Reviewer #2: N/A

4. Have the authors made all data underlying the findings in their manuscript fully available?

Reviewer #1: Yes

Reviewer #2: Yes

5. Is the manuscript presented in an intelligible fashion and written in standard English?

Reviewer #1: Yes

Reviewer #2: Yes

6. Review Comments to the Author

Reviewer #1: The manuscript is correct to be published after the revision. I consider that author/s has/have answered all the comments and the article merit publication.

Reviewer #2: García-Alaminos et al.

Great revision and I highly recommend the editor to publish this paper on PLOS ONE.

7. PLOS authors have the option to publish the peer review history of their article (what does this mean?). If published, this will include your full peer review and any attached files.

Reviewer #1: Yes: María del Carmen Valls Martínez

Reviewer #2: No

---

## [Editor Report · Acceptance letter]

3 Feb 2020

PONE-D-19-26281R1 

Unmasking social distant damage of developed regions’ lifestyle: A decoupling analysis of the indecent labour footprint 

Dear Dr. García-Alaminos:

I am pleased to inform you that your manuscript has been deemed suitable for publication in PLOS ONE. Congratulations! Your manuscript is now with our production department. 

With kind regards,

on behalf of

Dr. Jordi Paniagua 

Academic Editor

PLOS ONE